 

EMBO
Molecular Medicine

# Accumulated common variants in the broader fragile X gene family modulate autistic phenotypes

Beata Stepniak[1,†], Anne Kästner[1,2,†], Giulia Poggi[1,†], Marina Mitjans[1], Martin Begemann[1], Annette Hartmann[3], Sandra Van der Auwera[4], Farahnaz Sananbenesi[5], Dilja Krueger-Burg[6], Gabriela Matuszko[7], Cornelia Brosi[8], Georg Homuth[9], Henry Völzke[10], Fritz Benseler[6], Claudia Bagni[11,12], Utz Fischer[8], Alexander Dityatev[7], Hans-Jörgen Grabe[4], Dan Rujescu[3], Andre Fischer[5,13] & Hannelore Ehrenreich[1,2,*]

## Abstract

Fragile X syndrome (FXS) is mostly caused by a CGG triplet expansion in the fragile X mental retardation 1 gene (*FMR1*). Up to 60% of affected males fulfill criteria for autism spectrum disorder (ASD), making FXS the most frequent monogenetic cause of syndromic ASD. It is unknown, however, whether normal variants (independent of mutations) in the fragile X gene family (*FMR1, FXR1, FXR2*) and in *FMR2* modulate autistic features. Here, we report an accumulation model of 8 SNPs in these genes, associated with autistic traits in a discovery sample of male patients with schizophrenia (*N* = 692) and three independent replicate samples: patients with schizophrenia (*N* = 626), patients with other psychiatric diagnoses (*N* = 111) and a general population sample (*N* = 2005). For first mechanistic insight, we contrasted microRNA expression in peripheral blood mononuclear cells of selected extreme group subjects with high-versus low-risk constellation regarding the accumulation model. Thereby, the brain-expressed miR-181 species emerged as potential "umbrella regulator", with several seed matches across the fragile X gene family and *FMR2*. To conclude, normal variation in these genes contributes to the continuum of autistic phenotypes.

**Keywords** FMR1; FMR2; FXR1; FXR2; miR-181; PGAS
**Subject Categories** Genetics, Gene Therapy & Genetic Disease; Neuroscience

## Introduction

Fragile X syndrome (FXS) is associated with symptoms ranging from learning, motor and emotional deficiencies to mental retardation (IQ < 70) and autism (Garber *et al*, 2008). Up to 60% of males with FXS fulfill criteria for autism spectrum disorder (ASD) (Hagerman *et al*, 1986; Bailey *et al*, 1998; Clifford *et al*, 2007; Harris *et al*, 2008), making FXS the most common monogenetic cause of syndromic ASD (Hagerman *et al*, 2011). Almost all individuals with FXS show at least some autistic characteristics like social withdrawal (Hatton *et al*, 2006; Dahlhaus & El-Husseini, 2010; Heitzer *et al*, 2013). Since FXS is an X-linked disorder, males are generally more severely affected, with a suggested prevalence in Caucasians ranging from 1/3,717 to 1/8,918 (Crawford *et al*, 2001, 2002; Coffee *et al*, 2009).

FXS is in most cases caused by a CGG triplet expansion in the 5′-untranslated (UTR) region of the fragile X mental retardation 1 gene (*FMR1*). More than 200 repeat copies are considered a full mutation, triggering hypermethylation of the CpG island in the promoter region. This hypermethylation leads to transcriptional silencing of *FMR1* and loss of the associated protein, fragile X mental retardation protein (FMRP) (Oberle *et al*, 1991; Pieretti *et al*, 1991; Verkerk *et al*, 1991). FMRP is an RNA-binding protein, abundantly expressed in the mammalian brain, where it binds 4% of the whole transcriptome, including its own message (Ashley *et al*, 1993). Since FMRP interacts with many other proteins, its absence has manifold consequences—in sum affecting neural development,

1 Clinical Neuroscience, Max Planck Institute of Experimental Medicine, Göttingen, Germany
2 DFG Research Center for Nanoscale Microscopy and Molecular Physiology of the Brain (CNMPB), Göttingen, Germany
3 Department of Psychiatry and Psychotherapy, University of Halle, Halle, Germany
4 Department of Psychiatry and Psychotherapy, University Medicine Greifswald, Greifswald, Germany
5 Epigenetics in Neurodegenerative Diseases, German Center for Neurodegenerative Diseases (DZNE), Göttingen, Germany
6 Molecular Neurobiology, Max Planck Institute of Experimental Medicine, Göttingen, Germany
7 Molecular Neuroplasticity, German Center for Neurodegenerative Diseases (DZNE), Magdeburg, Germany
8 Department of Biochemistry, University of Würzburg, Würzburg, Germany
9 Interfaculty Institute for Genetics and Functional Genomics, University Medicine Greifswald, Greifswald, Germany
10 Institute for Community Medicine, University Medicine Greifswald, Greifswald, Germany
11 KU Leuven, Center for Human Genetics and Leuven Institute for Neurodegenerative Diseases, Leuven, Belgium
12 Department of Biomedicine and Prevention, University of Rome "Tor Vergata", Rome, Italy
13 Department of Psychiatry & Psychotherapy, University of Göttingen, Göttingen, Germany
*Corresponding author. Tel: +49 551 3899 628; Fax: +49 551 3899 670; E-mail: ehrenreich@em.mpg.de
†These authors contributed equally to this work

synapse formation, and plasticity (Bassell & Warren, 2008; Darnell *et al*, 2011; Pasciuto & Bagni, 2014a). A premutation syndrome (55–200 repeats) has also been reported with elevated *FMR1* mRNA and reduced FMRP levels, where RNA toxicity is a possible underlying molecular mechanism (Garcia-Arocena & Hagerman, 2010; Bagni *et al*, 2012). Premutation carriers display only subtle symptoms which are, however, still reminiscent of FXS, including deficits in social cognition, executive functioning, working memory, or selective attention (Moore *et al*, 2004; Cornish *et al*, 2005, 2008; Jacquemont *et al*, 2007; Kogan *et al*, 2008). Many of the FMRP mRNA targets, for example *CAMK2A*, *PSD-95*, *GABRB1*, *NLGN2*, have been linked to schizophrenia or ASD (Pasciuto & Bagni, 2014b). The most recent genomewide association study (GWAS) for schizophrenia described an enrichment of FMRP targets among the genomewide significant hits (Schizophrenia Working Group of the PGC, 2014), and in the largest whole exome sequencing study for schizophrenia, enhanced *de novo* mutations in mRNA targets of FMRP were found (Fromer *et al*, 2014).

Two autosomal homologues of *FMR1* exist—fragile X mental retardation autosomal homolog 1 (*FXR1*) and 2 (*FXR2*)—together forming the fragile X family of genes (Zhang *et al*, 1995). Both of these homologues encode also for RNA-binding proteins, FXR1P and FXR2P, respectively, with functions similar and complementary to FMRP (Penagarikano *et al*, 2007; Ascano *et al*, 2012). For instance, FMRP and FXR2P co-regulate crucial synaptic proteins like PSD95 (Fernandez *et al*, 2015). Interestingly, genomewide significant hits for schizophrenia also encompass the *FXR1* locus (Schizophrenia Working Group of the PGC, 2014).

Besides FXS, there is a phenotypically related unstable triplet expansion disorder, associated with mild mental retardation, the so-called fragile XE syndrome (Gecz, 2000). The mutation—similar to FXS—is due to an expansion of a CCG repeat beyond 200 in the 5′UTR of the AF4/FMR2 family member 2 (*AFF2,* also called *FMR2*), which leads to hypermethylation of the CpG island upstream of *FMR2* and transcriptional gene silencing (Knight *et al*, 1993; Gecz *et al*, 1996; Gu *et al*, 1996). FMR2 is a nuclear protein expressed in fetal and adult brain and belongs to a gene family of transcription activators (Gecz *et al*, 1997; Hillman & Gecz, 2001). Importantly, an increased number of missense mutations in *FMR2* was found in male patients with ASD compared to controls (Mondal *et al*, 2012).

In summary, there seems to be a considerable connection of both schizophrenia and ASD with the "broader fragile X family" of genes, in which we have included *FMR2* based on the striking functional/phenotypical similarities and interactions. Along with the genetic relationship between these mental disorders, clinical overlaps have also been described, such as the shared impairment of specific cognitive domains like theory of mind (King & Lord, 2011; Owen *et al*, 2011).

Surprisingly, all that is known about genotype–phenotype associations in this broader fragile X gene family is derived from mutations, but the contribution of common, frequent variations in these genes to the normal continuum of autism-related phenotypes, for example social interaction, communication, or stereotypies, has never been investigated. In the present study, we asked for the first time whether accumulated common genetic variants in genes of the broader fragile X gene family (*FMR1, FXR1, FXR2, FMR2*) modulate autistic features in males, independently of the described mutations,

that is, the polymorphic repeats in *FMR1* and *FMR2*. For quantification of autistic phenotypes, we used the PAUSS, an autism severity score composed of specific items of the Positive and Negative Syndrome Scale (PANSS autism severity score) (Kästner *et al*, 2015).

We report here an accumulation model of 8 single nucleotide polymorphisms (SNPs) that yields significant association with autistic traits in a schizophrenic discovery and two independent replicate samples of mentally ill subjects, as well as one replicate sample from the general population. In a first and still preliminary approach toward mechanistic insight, we employed small RNA sequencing and found lower expression of miR-181 species in peripheral blood mononuclear cells (PBMC) of subjects with high- versus low-risk constellation in the 8-SNP model. The fact that this microRNA family has several seed matches across the broader fragile X gene family may suggest an overarching regulatory mechanism.

## Results

### Length distribution of *FMR1* and *FMR2* repeat polymorphisms in the male schizophrenic discovery sample is indistinguishable from healthy individuals

As prerequisite for exploring the contribution of normal variation in genes of the broader fragile X gene family (*FMR1*, *FXR1*, *FXR2*, *FMR2*) to the overall continuum of autism-related phenotypes, we had to determine the polymorphic repeat lengths in *FMR1* and *FMR2* in the schizophrenic discovery and the healthy control sample to exclude mutations. As illustrated in Fig 1, repeats were similarly distributed in the Göttingen Research Association for Schizophrenia (GRAS) patients and healthy controls. All had < 50 CGG and < 40 CCG repeats in *FMR1* and *FMR2*, respectively, that is, far away even from premutation carrier status (Tassone *et al*, 2014). We then checked whether the (normal) repeat length would still have any relevance for schizophrenia symptom severity in the discovery sample. As shown in Table 1, no associations were found with age at disease onset, positive, cognitive, neurological symptoms or PAUSS (Kästner *et al*, 2015). Even comparing the top and bottom 10% of GRAS individuals with smallest or largest repeat lengths did not result in any significant differences (Table 1).

### An accumulation model of 8 proautistic genotypes of the broader fragile X family predicts autistic phenotypes in the schizophrenia discovery sample

Having a comparable basis of repeat polymorphism distribution in the schizophrenia discovery sample with no obvious influence on disease readouts, we first selected SNPs in the broader fragile X gene family according to our standard operating procedure (SOP) as meticulously described in Figs 2 and 3 and in the materials and methods section. The high internal consistency of all individual PAUSS items (Kästner *et al*, 2015) allowed their aggregation to form a single dimensional measure of the severity of autistic symptoms (Fig 4A) and to explore the preselected 13 SNPs (Fig 2) individually with respect to potential proautistic genotypes (SOP: Fig 3). According to this SOP, the following 8 proautistic genotypes out of the 13

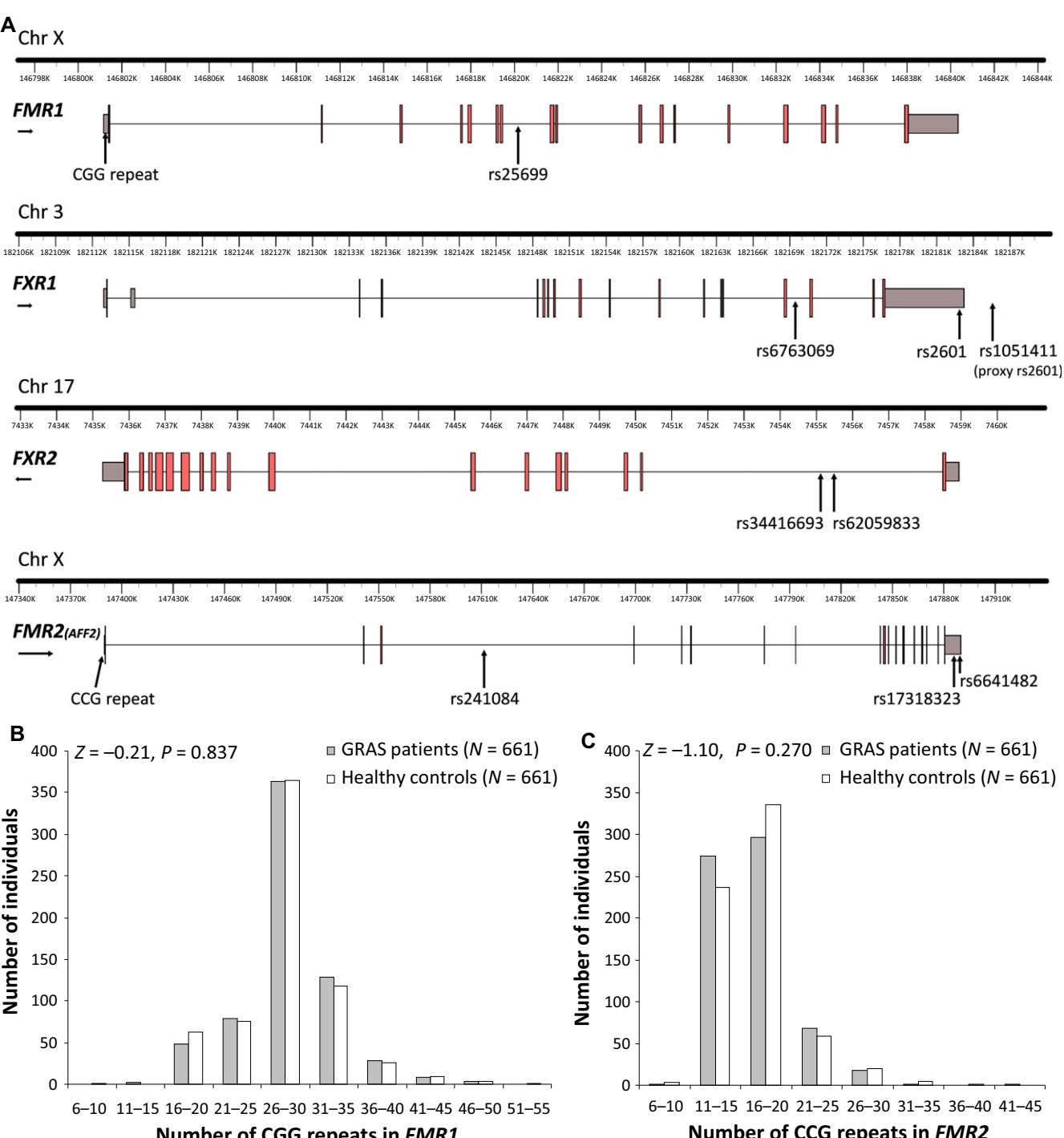

**Figure 1. Positions of SNPs in *FMR1*, *FXR1*, *FXR2*, and *FMR2*, forming the 8-SNP model as well as *FMR1* and *FMR2* repeat polymorphism length distribution in the discovery sample.**

A   Schematic overview of *FMR1*, *FXR1*, *FXR2*, *FMR2*, and position of the 8 selected single nucleotide polymorphisms (SNPs). Line represents introns, gray box at the beginning and end of each gene stands for UTR region, and red boxes represent exons. Gene structure plots generated using FancyGene (Rambaldi & Ciccarelli, 2009).

B   Distribution of repeat polymorphism lengths in *FMR1* of male GRAS schizophrenia patients and healthy controls.

C   Distribution of repeat polymorphism lengths in *FMR2* of male GRAS schizophrenia patients and healthy controls.

SNPs were chosen that revealed a tendency of a higher PAUSS (i.e. higher severity of autistic symptoms): T in *FMR1* rs25699, TT in *FXR1* rs6763069, AA in *FXR1* rs2601, GG in *FXR2* rs34416693, CC in *FXR2* rs62059833, A in *FMR2* rs241084, G in *FMR2* rs17318323, and G in *FMR2* rs6641482. These genotypes were used for generation of the 8-SNP accumulation model.

**Table 1.** Correlation of repeat length polymorphisms with measures of schizophrenia disease severity and autistic features in the male schizophrenia GRAS sample and extreme group comparison of repeat length polymorphisms for the same measures.

| | FMR1 repeat polymorphism | | | | | FMR2 repeat polymorphism | | | | |
|---|---|---|---|---|---|---|---|---|---|---|
| | Correlation coefficient $n = 596-654^a$ | P-value | 10% with shortest repeats (mean ± SD) $n = 68-70^a$ | 10% with longest repeats (mean ± SD) $n = 58-67^a$ | Z, T value P-value | Correlation coefficient $n = 595-654^a$ | P-value | 10% with shortest repeats (mean ± SD) $n = 63-70^a$ | 10% with longest repeats (mean ± SD) $n = 64-70^a$ | Z, T value P-value |
| Age at disease onset | $r_s = 0.024$ | $P = 0.540$ | 24.75 ± 7.95 | 24.25 ± 7.25 | $Z = -0.36$ $P = 0.721$ | −0.040 | $P = 0.313$ | 24.16 ± 8.46 | 24.18 ± 7.57 | $Z = -0.27$ $P = 0.787$ |
| PANSS positive | $r_s = -0.057$ | $P = 0.149$ | 14.24 ± 5.70 | 13.20 ± 5.62 | $Z = -1.11$ $P = 0.269$ | −0.016 | $P = 0.678$ | 14.37 ± 6.37 | 13.45 ± 5.80 | $Z = -0.94$ $P = 0.350$ |
| Cognitive composite score[b] | $r_s = 0.021$ | $P = 0.595$ | 0.07 ± 1.12 | −0.07 ± 0.97 | $t = 0.70$ $P = 0.484$ | 0.035 | $P = 0.378$ | 0.02 ± 0.99 | 0.04 ± 0.96 | $t = -0.08$ $P = 0.935$ |
| CNI[b] | $r_s = -0.037$ | $P = 0.362$ | −0.02 ± 0.89 | 0.03 ± 0.95 | $Z = -0.34$ $P = 0.736$ | 0.035 | $P = 0.400$ | −0.05 ± 0.84 | 0.04 ± 0.91 | $Z = -0.69$ $P = 0.494$ |
| PAUSS[c] | $r_s = 0.006$ | $P = 0.870$ | −0.03 ± 0.67 | 0.05 ± 0.75 | $Z = -0.53$ $P = 0.598$ | −0.027 | $P = 0.499$ | 0.06 ± 0.64 | −0.03 ± 0.61 | $Z = -0.67$ $P = 0.504$ |

Uncorrected mean ± SD presented. Spearman's correlation coefficient ($r_s$) was calculated for *FMR1/FMR2* repeat polymorphisms and respective disease measures. For further statistical analysis of extreme groups (10% with longest and 10% with shortest repeats), Mann–Whitney *U*-test or *t*-test for normally distributed variables was used. PANSS, positive and negative syndrome scale; CNI, Cambridge neurological inventory.
[a]Because of missing data, sample sizes vary.
[b]Corrected for age and chlorpromazine equivalents (standardized residual after linear regression).
[c]Z-standardized PANSS autism severity score.

To exclude that any of the so-selected 8 SNPs would be associated with the diagnosis of schizophrenia, we first conducted a case–control analysis of the male GRAS schizophrenic and healthy control subjects, yielding no statistically significant results. All markers fulfill Hardy–Weinberg equilibrium after significance level was corrected for multiple testing ($P < 0.013$) (Table 2). Hence, all individuals of the discovery sample could now be ranked according to their number of proautistic genotypes. In this sense, for the autosomal genes (*FXR1*, *FXR2*), homozygous proautistic genotypes were always counted as 1; heterozygous proautistic genotypes as 0.5 and the non-proautistic genotypes as 0. For the X-chromosomal genes (*FMR1*, *FMR2*), the proautistic genotypes were counted as 1 and the non-proautistic genotypes as 0 (Fig 3). Figure 4B displays the average PAUSS of all individuals in the discovery sample dependent on the number of accumulated proautistic genotypes. Higher numbers correlate with higher PAUSS ($r_s = 0.103$, $P = 0.008$). In contrast, these numbers do not correlate with unrelated control phenotypes, for example, our delusional depression composite score (Fig 3C and D).

When contrasting extreme group individuals, that is, those with 1.5–2.5 to those with 5.5–6.5 proautistic genotypes, a highly significant difference ($P = 0.0002$) regarding severity of autistic features emerged (Fig 4C). Table 3 highlights the overall contrast between the two extreme groups with regard to autism-relevant measures, that is, PAUSS and individual score items, with the most striking group difference of all single items seen for social withdrawal. In contrast, none of the control variables differs between the extreme groups, including lengths of repeat polymorphisms, age, positive, cognitive, or neurological symptom severity (Table 3).

As internal validity control of the 8-SNP model, an exploratory exclusion of each SNP was performed to learn whether the significance level of the model's associations with PAUSS or its subitems as presented in Table 3 would be affected. In all 8 of the so-created

7-SNP models, essentially all associations deteriorated, thus clearly supporting the chosen 8-SNP model (Table 4).

**The association of the 8-SNP model derived from the broader fragile X gene family with autistic phenotypes is replicated in three independent samples**

To test whether the associations of the 8-SNP model with autistic phenotypes found in the GRAS discovery sample were consistent, we employed three independent replication samples: (I) male patients with schizophrenia from Munich/Halle ($N = 626$), (II) male patients with other psychiatric diagnoses (extended GRAS data collection; $N = 111$), and (III) a general population sample of males from Greifswald ($N = 2{,}005$). For replication samples I and II, the PAUSS was employed, again resulting in clear differences between extreme groups with low and high numbers of proautistic genotypes, comparable to the discovery sample (Fig 4D and E). For replication sample III, social support [derived from the Instrumental Support Index (Klein *et al*, 2012)] was used as a proxy phenotype. Reassuringly, in the discovery sample, social support (operationalized as the self-reported number of individuals a person can rely on in case of emergency) correlated substantially with the PAUSS (Spearman rank correlation for $N = 639$: $r_s = -0.313$, $P = 5 \times 10^{-16}$, Fig 4F), underlining the relevance of this phenotype for autism. For the PAUSS item social withdrawal, the correlation was even more pronounced than for the overall PAUSS (Spearman rank correlation for $N = 649$: $r_s = -0.337$, $P = 1.2 \times 10^{-18}$). Extreme groups with low and high numbers of proautistic genotypes diverged substantially regarding social support in the discovery sample as well as replication sample III (Fig 4G and H). To conclude, in all three replication samples, we confirmed the association of the 8-SNP model, derived from the broader fragile X gene family (*FMR1*, *FXR1*, *FXR2*, *FMR2*), with autism-related behaviors,

**A**

| FMR1 Xq27.3 | FXR1 3q26.33 | FXR2 17p13.1 | FMR2 Xq28 |

**B**

10 SNPs available     13 SNPs available     5 SNPs available     152 SNPs available

**C**

| | | | |
|---|---|---|---|
| 1 functional & MAF | 1 functional & MAF | 2 MAF & LD | 1 functional & MAF |
| 1 MAF & LD | 3 MAF & LD | | 3 functional & LD |
| 2 MAF | 4 MAF | | 1 functional |
| 5 LD | 5 LD | | 16 MAF & LD |
| | | | 97 MAF |
| | | | 16 LD |

8 SNPs included in the accumulation model

**D**

| rs25699 (MAF & LD) | rs2601 (functional & MAF) | rs34416693 (MAF & LD) | rs241084 (functional & MAF) |
| | rs6763069 (MAF & LD) | rs62059833 (MAF & LD) | rs17318323 (functional & LD) |
| | | | rs6641482 (functional & LD) |

**Figure 2. SNP overview and unbiased selection according to the standard operating procedure (SOP) developed for phenotype-based genetic association study (PGAS) approaches.**

A  Genes from the "broader fragile X family" and their chromosomal position.

B  SNP numbers available through direct genotyping in the here used semicustom genotyping array (Affymetrix, Santa Clara, CA, USA).

C  SNPs fulfilling some of the first round of selection criteria ("functional" = SNPs, i.e. located in promoter region, 3′UTR or coding sequence; MAF = MAF ≥ 0.2; LD = SNPs that "survive" after linkage disequilibrium pruning: $r^2 < 0.8$). Underlined are the 13 SNPs selected for the PGAS approach using the PAUSS (selection requirements: fulfilled 2 of the above criteria or were functional). Not more than 3 SNPs per gene are selected to avoid overrepresentation of one gene.

D  SNPs with a tendency in PGAS (see Fig 3A) at single SNP basis went into the final accumulation model.

underlining the phenotypical continuum of these traits across health and disease.

**Autism-relevant 8-SNP model derived from the broader fragile X gene family: Toward first mechanistic insight**

Next, the obvious question arose of how to explain the role in aggregate of variants within the broader fragile X gene family for the development of autistic features. Interactions and/or partial compensation/substitution of some of these genes for each other have already been suggested [for example Ceman *et al* (1999), Jin *et al* (2004), Fernandez *et al* (2015)]. In a preliminary approach toward mechanistic insight, we chose the functionally interesting SNP rs2601 A/G, located in the 3′UTR of *FXR1. In silico* prediction using PITA algorithm (Kertesz *et al*, 2007) revealed here an allele dependent variable binding of miR-181 species in case of G versus A. For the G allele, the ΔΔG prediction ranges from −3.84 to −9.90; for the A allele, it is 0 (AA is the proautistic risk genotype in our model) (Fig 5). Not only brain, but also thymus or PBMC express relatively high amounts of miR-181 species (Hsu *et al*, 2006; Landgraf *et al*, 2007; Asquith *et al*, 2014). Therefore, we hypothesized that *FXR1* mRNA expression in PBMC should be lower in GG carriers due to the predicted strong binding of miR-181 species. Comparing *FXR1* expression in subjects homozygous for rs2601 G ($N = 16$) versus A ($N = 27$), we saw a tendency in the expected direction (Fig 5A).

We now wondered whether the other seven risk genotypes in the model would contribute to alterations in the regulatory microenvironment, for example, the microRNAome. For this, we controlled for the effect of the rs2601 genotype by selecting only AA carriers. We formed two extreme groups ($N = 6$ each), that is, with 2–2.5 versus 5.5–6.5 proautistic genotypes now also considering the remaining 7 SNPs. Contrasting microRNA expression in PBMC of these 2 extreme groups with high versus low autism score/genetic risk, we found again miR-181 emerge. In order to quantify all miR-181 molecules that target the broader fragile X gene family, we performed small RNA sequencing and saw that all miR-181 species (a, b, c, d-5p) together were lower expressed in high-risk subjects ($P = 0.024$) (Fig 5B). We next had a closer look at the miR-181 family members. Figure 5C illustrates the sequence homology of all four mature human miR-181 species and the remarkable number of miR-181 seed matches in the broader fragile X gene family. Only in *FXR2*, no matching sequence was found (Fig 5C). Figure 5D gives the predicted ΔΔG for each of the miR-181 family members within different 3′UTR positions in the broader fragile X gene family. Positions were identified using Target Scan Human (http://www.targetscan.org) and SFOLD (http://sfold.wadsworth.org/cgi-bin/index.pl) and then processed using PITA algorithm (Kertesz *et al*, 2007) to yield the denoted ΔΔG values.

In contrast to the differentially expressed microRNAs in PBMC of our extreme group subjects, FMRP levels were found to be similar upon quantification by Western blot [1.5–2.5 ($N = 6$) versus 5.5–6.5 ($N = 7$) risk genotypes: 0.418 ± 0.493 versus 0.522 ± 0.651, relative units mean ± SEM; $P = 0.754$].

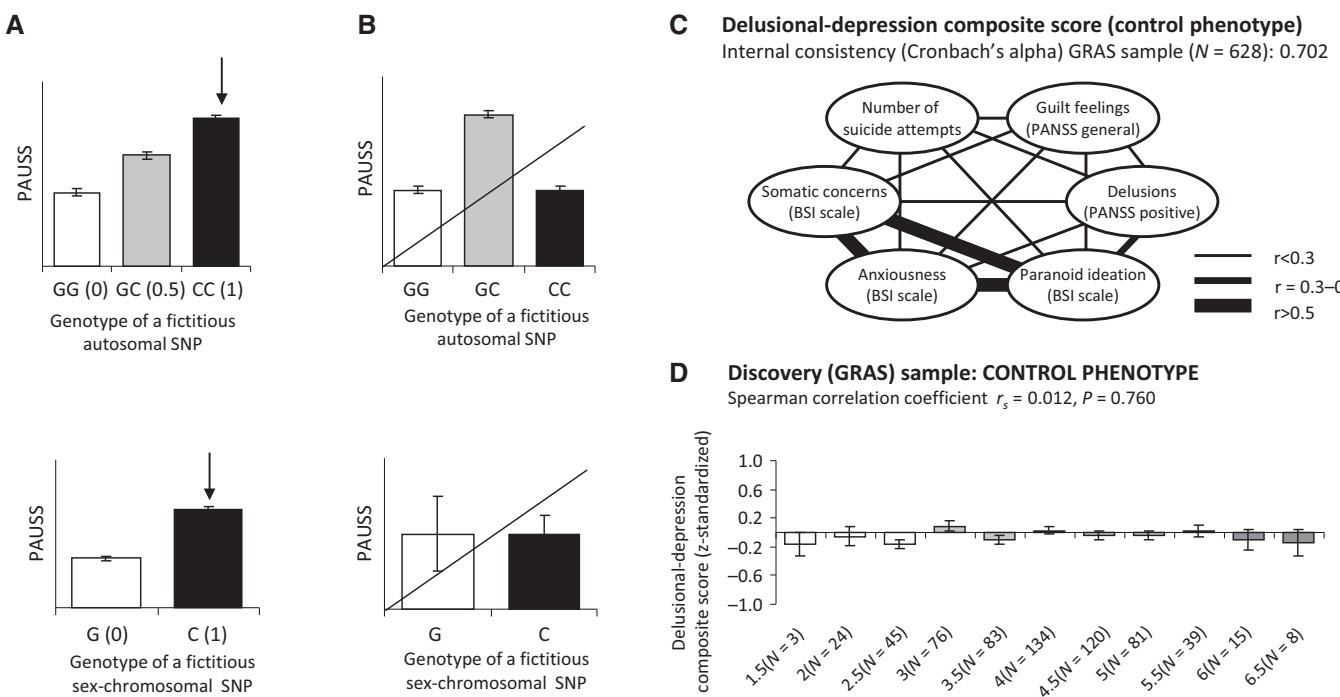

**Figure 3.  Criteria of final SNP selection in the phenotype-based genetic association study (PGAS) approach—Standard operating procedure (SOP).**

A, B   A total of 13 SNPs preselected according to the SOP presented in Fig 2 underwent PGAS screening as exemplified here: (A) PAUSS association pattern of an exemplary fictitious autosomal (upper panel) and a sex-chromosomal (lower panel) SNP which are eligible for the accumulation model. The genotype associated with the highest average PAUSS (in this example CC) is the "proautistic genotype" (indicated by the black arrow) and is assigned a score of 1. The heterozygous genotype is assigned a score of 0.5 and the homozygous genotype associated with the lowest PAUSS receives a score of 0. Please note that the difference between genotypes does not have to be statistically significant. (B) PAUSS association pattern of an exemplary fictitious autosomal (upper panel) and a sex-chromosomal (lower panel) SNP which would not be selected for the accumulation model because of unclear phenotypical/biological relevance.

C, D   The specificity of the association of the 8-SNP accumulation model with an autistic phenotype (as determined using PAUSS; compare Fig 4B) is controlled by applying an unrelated (or "non-sense") phenotype, for example, delusional-depression: (C) Intercorrelation pattern of single items included in the delusional-depression composite score, used here as example control phenotype. Cronbach's alpha is presented as measure of internal consistency. (D) The delusional-depression composite score is not associated with the number of proautistic genotypes of the 8-SNP risk model in the discovery sample.

Data information: Error bars represent SEM.

## Discussion

In the present study, we provide a model of 8 common genetic variants in genes of the broader fragile X family that co-modulate autistic traits in a discovery sample of male patients with schizophrenia and in three independent replication samples, comprising: (I) schizophrenic individuals, (II) other psychiatric diagnoses, and (III) general population. This co-modulation is independent of the known mutations, that is, *FMR1* and *FMR2* repeat polymorphism lengths. Extreme groups carrying high versus low numbers of the 8-SNP model of risk genotypes do not only differ substantially regarding the severity of autistic phenotypes but apparently also with respect to the microRNAome as determined in PBMC.

We have termed the kind of approach taken here PGAS (phenotype-based genetic association study) which allows elucidating the contribution of normal genetic variation to (disease) phenotypes, and thereby ultimately aims at re-defining disease entities based on biological grounds (Ehrenreich & Nave, 2014). The results of the present PGAS nicely illustrate that common genetic variants

in aggregate can at least co-determine a psychiatric disease phenotype and that mutations may not necessarily be required in all cases. In fact, non-syndromic ASD is estimated to be in 10–20% monogenic, that is, caused by a clear-cut genetic mutation (Geschwind, 2011). Most cases are etiologically unclear. Some of them could well be derived from an unfortunate aggregation of normal genotypes as shown here, possibly combined with environmental risk factors in the sense of a second hit (Tordjman *et al*, 2014).

For quantification of autistic traits, we employed our novel, easy-to-apply dimensional PANSS-based autism severity score (PAUSS) which has previously been cross-validated with the autism diagnostic observation schedule (ADOS), an established autism-scoring instrument (Kästner *et al*, 2015). Importantly, we did not find the 8-SNP model to be associated with any other readout of schizophrenia disease severity or control phenotypes, emphasizing its specific role for autistic traits. In particular, social withdrawal seemed to be substantially influenced by the number of proautistic genotypes in the broader fragile X gene family, an observation in good agreement with the literature on autistic features in fragile X (and related) gene mutation carriers, man and

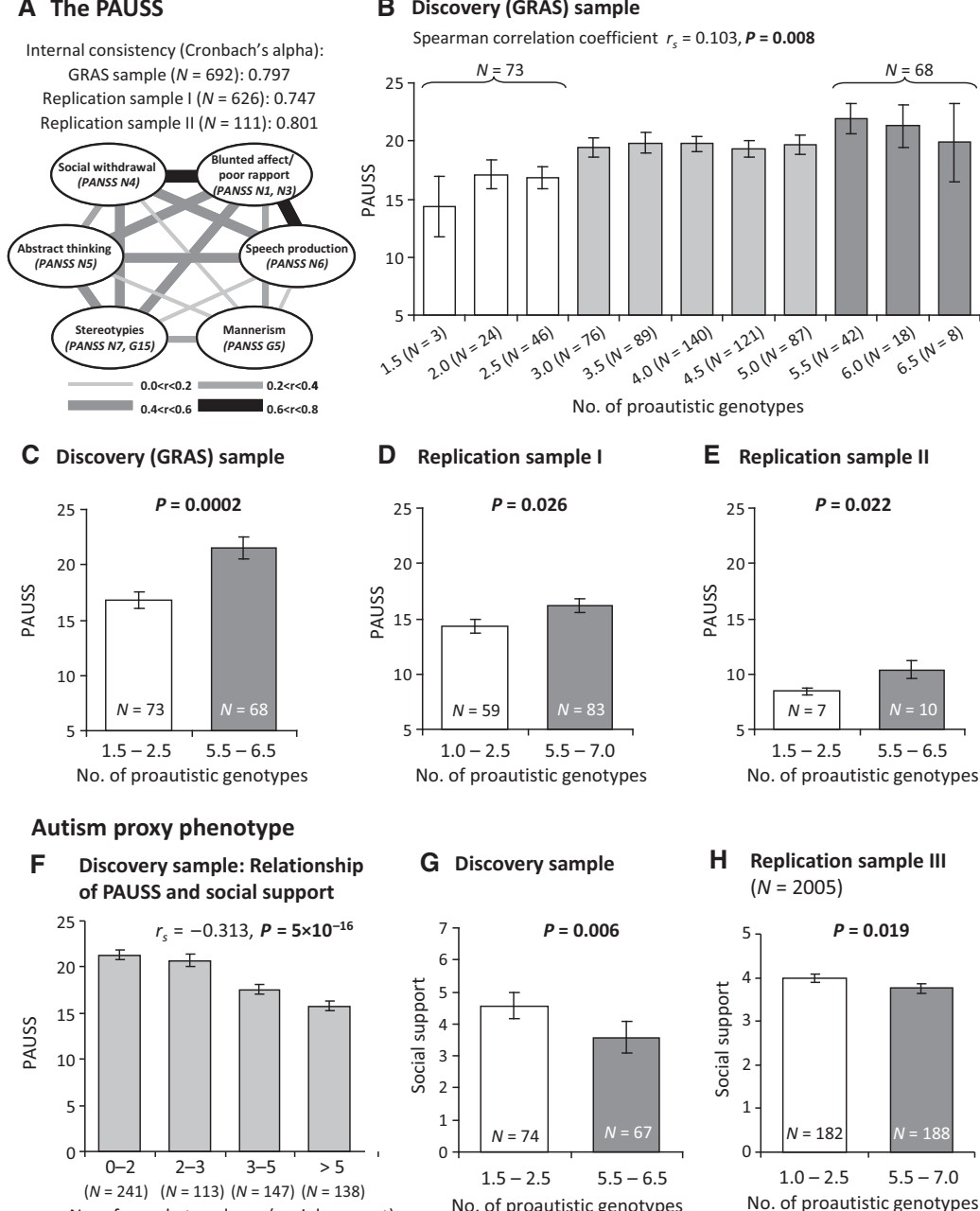

**Figure 4. Association of autism severity readouts in the discovery and 3 independent replication samples with the number of proautistic genotypes in the 8-SNP risk model derived from the broader fragile X gene family.**

A   PAUSS (PANSS autism severity score) composition and item intercorrelation pattern in the GRAS sample of male schizophrenic individuals (discovery sample). Cronbach's alpha is presented as measure of internal consistency and also provided for the male replication samples I and II.

B   Association of PAUSS with the number of proautistic genotypes of the 8-SNP risk model in the discovery sample; mean ± SEM.

C   PAUSS comparison of extreme groups with high and low numbers of accumulated proautistic genotypes in the discovery sample; binary logistic regression analysis with non-z-standardized PAUSS as dependent variable; mean ± SEM.

D   PAUSS comparison of extreme groups with high and low numbers of accumulated proautistic genotypes in replication sample I of male schizophrenia patients; binary logistic regression analysis; mean ± SEM.

E   PAUSS comparison of extreme groups with high and low numbers of accumulated proautistic genotypes in replication sample II of male disease control patients; Mann–Whitney *U*-test; mean ± SEM.

F   The highly significant correlation of PAUSS and social support underlines the validity of social support as an autism proxy phenotype; mean ± SEM.

G   Comparison of the extreme groups with high and low numbers of accumulated proautistic genotypes for the autism proxy phenotype social support in the discovery sample; binary logistic regression analysis; mean ± SEM.

H   Comparison of the extreme groups with high and low numbers of accumulated proautistic genotypes for the autism proxy phenotype social support in the male replication sample III from general population; binary logistic regression analysis; mean ± SEM. For all replications, *P*-values for one-sided tests are shown.

**Table 2.  Hardy–Weinberg statistics and case–control analysis for male patients with schizophrenia and healthy controls (GRAS).**

**(A)**

| | | Test for HWE deviation | | |
|---|---|---|---|---|
| Gene | SNP | *P*-value total | *P*-value controls | *P*-value cases |
| *FXR1* | rs6763069 | 0.132 | 0.933 | 0.038 |
| *FXR1* | rs2601 | 0.757 | 0.593 | 0.911 |
| *FXR2* | rs34416693 | 0.116 | 0.358 | 0.653 |
| *FXR2* | rs62059833 | 0.089 | 0.015 | 0.924 |

**(B)**

| | | | | *N* (%) | | |
|---|---|---|---|---|---|---|
| Gene | SNP | MAF | Genotype | Cases (*N* = 709) | Controls (*N* = 730) | *P*-value |
| *FMR1* | rs25699 | 0.416 | T | 412 (58.3) | 455 (62.7) | 0.091 |
| | | | C | 294 (41.7) | 271 (37.3) | |
| *FXR1* | rs6763069 | 0.325 | AA | 331 (47.4) | 319 (44.5) | 0.872 |
| | | | AT | 282 (40.3) | 317 (44.3) | |
| | | | TT | 86 (12.3) | 80 (11.2) | |
| *FXR1* | rs2601 | 0.212 | AA | 435 (61.7) | 438 (60.2) | 0.428 |
| | | | AG | 237 (33.6) | 257 (35.3) | |
| | | | GG | 33 (4.7) | 33 (4.5) | |
| *FXR2* | rs34416693 | 0.302 | GG | 342 (49.3) | 368 (50.8) | 0.280 |
| | | | GA | 287 (41.3) | 304 (42.0) | |
| | | | AA | 65 (9.4) | 52 (7.2) | |
| *FXR2* | rs62059833 | 0.278 | CC | 368 (53.0) | 388 (53.6) | 0.307 |
| | | | CT | 274 (39.5) | 301 (41.6) | |
| | | | TT | 52 (7.5) | 35 (4.8) | |
| *FMR2* | rs241084 | 0.272 | A | 514 (73.6) | 544 (75.6) | 0.393 |
| | | | G | 185 (26.5) | 176 (24.4) | |
| *FMR2* | rs17318323 | 0.069 | A | 660 (93.2) | 687 (94.1) | 0.505 |
| | | | G | 48 (6.8) | 43 (5.9) | |
| *FMR2* | rs6641482 | 0.112 | A | 629 (88.7) | 648 (88.9) | 0.887 |
| | | | G | 80 (11.3) | 81 (11.1) | |

(A) Test for deviation from Hardy–Weinberg equilibrium (HWE) in male schizophrenic individuals (GRAS) and male healthy controls for non-X-chromosomal SNP genotypes; significance level after correction for multiple testing *P* < 0.013. (B) No significant differences in genotype frequencies in case–control comparison of male schizophrenic patients (GRAS) with healthy controls.

**Table 3.  Comparison of extreme groups with high and low numbers of accumulated proautistic genotypes in the male GRAS sample regarding autism-relevant measures and control variables.**

| | 1.5–2.5 proautistic genotypes *N* = 69–73[a] | 5.5–6.5 proautistic genotypes *N* = 57–68[a] | *Z, T* value *P*-value |
|---|---|---|---|
| Autism-relevant measures (mean ± SD) | | | |
| Social withdrawal | 2.10 ± 1.22 | 3.10 ± 1.67 | *Z* = −3.71 **P = 0.0002** |
| Mannerism | 1.27 ± 0.79 | 1.68 ± 1.22 | *Z* = −2.88 **P = 0.003** |
| Blunted affect & poor rapport | 2.53 ± 1.07 | 3.16 ± 1.38 | *Z* = −2.80 **P = 0.005** |
| Speech production | 1.81 ± 1.20 | 2.49 ± 1.48 | *Z* = −2.84 **P = 0.003** |
| Stereotyped thinking & preoccupation | 1.99 ± 1.04 | 2.48 ± 1.13 | *Z* = −2.84 **P = 0.005** |
| Difficulties in abstract thinking | 2.58 ± 1.59 | 2.96 ± 1.62 | *Z* = −1.45 P = 0.146 |
| PAUSS[b] | −0.22 ± 0.52 | 0.21 ± 0.74 | *Z* = −3.69 **P = 0.0002** |
| Control variables (mean ± SD) | | | |
| *FMR1* repeat length | 29.33 ± 4.79 | 28.90 ± 4.63 | *Z* = −1.44 P = 0.151 |
| *FMR2* repeat length | 17.41 ± 3.10 | 17.89 ± 3.72 | *Z* = −0.41 P = 0.681 |
| Age | 36.97 ± 10.79 | 38.74 ± 14.01 | *Z* = −0.69 P = 0.488 |
| PANSS positive | 11.99 ± 4.77 | 13.70 ± 6.41 | *Z* = −1.39 P = 0.165 |
| Cognitive composite score[c] | −0.07 ± 0.84 | −0.08 ± 1.02 | *T* = 0.44 P = 0.661 |
| CNI[c] | 0.06 ± 1.06 | 0.14 ± 0.95 | *T* = −0.58 P = 0.562 |

Statistically significant *P*-values are set in boldface (Bonferroni-corrected significance threshold: *P* < 0.007).
Mean ± SD presented. For statistical analysis, Mann–Whitney *U*-test or *t*-test for normally distributed variables was used. PANSS, positive and negative syndrome scale; CNI, Cambridge Neurological Inventory.
[a]Because of missing data, sample sizes vary.
[b]*Z*-standardized score.
[c]Corrected for age and chlorpromazine equivalents (standardized residual after linear regression).

mouse (Hatton *et al*, 2006; Dahlhaus & El-Husseini, 2010; Heitzer *et al*, 2013).

We directly replicated the association of increasing genetic risk in the 8-SNP accumulation model with increasing scores of PAUSS and its sub-items in schizophrenic subjects (replication sample I) and otherwise psychiatrically ill patients (replication sample II), even though in the latter sample on a much lower level of symptom severity (PAUSS) compared to the schizophrenia discovery and replication sample I. This fact is likely related to the disease spectrum included in replication II, mainly consisting of affective disorders and drug addiction. In these conditions, much less pronounced autistic phenotypes would be expected, even considering the general continuum of autistic traits in humans (Kästner *et al*, 2015). Finally, in line with the strong associations of the 8-SNP model obtained here

with social withdrawal as key feature of ASD, we were able to demonstrate that an accumulation of ≥ 5.5 proautistic genotypes, compared to ≤ 2.5 in a general population sample, is associated with less self-reported social support. These findings further underline the view that autistic traits can be found across different diagnostic groups and even among individuals of the general population as a phenotypic continuum and not just as a dichotomous category (present/absent).

Since 3 out of the 4 genes in the broader fragile X gene family are putative targets of miR-181 species, we speculate that this microRNA family may play an important role as an "umbrella regulator" of ASD-relevant genes. The term "umbrella regulator" is used here to describe a potential regulatory principle that is common to different independent genes (which are even located on

Table 4.  Change of *P*-values upon exclusion of each SNP from the original 8-SNP accumulation model – internal control.

| | PAUSS | Blunted affect | Poor rapport | Social withdrawal | Difficulties in abstract thinking | Speech production | Stereotyped thinking | Mannerism | Preoccupation |
|---|---|---|---|---|---|---|---|---|---|
| **8-SNP model** (1.5–2.5 versus 5.5–6.5 proautistic genotypes) (*N* = 69–73 versus *N* = 57–68)[a] | ***P* = 0.0002** | ***P* = 0.020** | ***P* = 0.002** | ***P* = 0.0002** | *P* = 0.146 | ***P* = 0.003** | *P* = 0.278 | ***P* = 0.003** | ***P* = 0.001** |
| SNP taken out of the 8-SNP accumulation model[a] | | | | | | | | | |
| rs25699 (*FMR1*) (*N* = 64 versus *N* = 11)[b] | *P* = 0.203 | *P* = 0.490 | *P* = 0.766 | *P* = 0.310 | *P* = 0.424 | *P* = 0.114 | *P* = 0.039[#] | *P* = 0.089 | *P* = 0.611 |
| rs6763069 (*FXR1*) (*N* = 48 versus *N* = 24–26) | *P* = 0.195 | *P* = 0.465 | *P* = 0.135 | *P* = 0.181 | *P* = 0.455 | *P* = 0.047 | *P* = 0.242 | *P* = 0.249 | *P* = 0.120 |
| rs2601 (*FXR1*) (*N* = 107 versus *N* = 10) | *P* = 0.595 | *P* = 0.594 | *P* = 0.709 | *P* = 0.358 | *P* = 0.899 | *P* = 0.131 | *P* = 0.239 | *P* = 0.520 | *P* = 0.955 |
| rs34416693 (*FXR2*) (*N* = 106 versus *N* = 18) | *P* = 0.800 | *P* = 0.766 | *P* = 0.385 | *P* = 0.376 | *P* = 0.625 | *P* = 0.032 | *P* = 0.403 | *P* = 0.325 | *P* = 0.750 |
| rs62059833 (*FXR2*) (*N* = 110 versus *N* = 16) | *P* = 0.075 | *P* = 0.342 | *P* = 0.291 | *P* = 0.038 | *P* = 0.306 | *P* = 0.007 | *P* = 0.529 | *P* = 0.610 | *P* = 0.528 |
| rs241084 (*FMR2*) (*N* = 95 versus *N* = 10) | *P* = 0.226 | *P* = 0.875 | *P* = 0.986 | *P* = 0.067 | *P* = 0.902 | *P* = 0.529 | *P* = 0.187[#] | *P* = 0.122 | *P* = 0.524 |
| rs17318323 (*FMR2*) (*N* = 31 versus *N* = 49) | *P* = 0.008 | *P* = 0.039 | *P* = 0.077 | *P* = 0.026 | *P* = 0.230 | *P* = 0.419 | *P* = 0.127[#] | *P* = 0.048 | *P* = 0.005 |
| rs6641482 (*FMR2*) (*N* = 38 versus *N* = 48) | *P* = 0.118 | *P* = 0.089 | *P* = 0.072 | *P* = 0.119 | *P* = 0.882 | *P* = 0.259 | *P* = 0.955 | *P* = 0.144 | *P* = 0.020 |

*P*-values ≤ 0.05 in the 8-SNP accumulation model are set in boldface.
[a]Individuals with 1.5–2.5 proautistic genotypes were compared to individuals with 5.5–6.5 proautistic genotypes by Mann–Whitney *U*-test.
[b]*N* numbers refer to the extreme groups (1.5–2.5 compared to 5.5–6.5 proautistic genotypes).
[#]*P*-values improve upon exclusion of the respective SNP only for 3 SNPs regarding stereotyped thinking; for all other SNPs and variables, *P*-values worsen upon exclusion of the respective SNP.

different chromosomes) that all carry seed sequences of the miR-181 species. This microRNA family might therefore influence their expression in the sense of an overarching regulatory mechanism. A principle risk constellation may be given if miR-181 binding is reduced (as in the 3′UTR SNP rs2601 AA), or if miR-181 levels are decreased (expression diminished or consumption/degradation increased), together resulting as net effect in less efficient downregulation of target genes (the latter refers to the remaining 7 SNPs of the model where high risk was associated with reduction of total miR-181).

The miR-181 family members identified in our pilot approach toward mechanistic insight are strongly brain-expressed. These multiple regulators have been previously associated with neurodevelopment, learning and memory function, glutamate signaling, and neuroinflammation, and even some preliminary hints were reported regarding autism, for example (Hutchison *et al*, 2013; Mundalil Vasu *et al*, 2014). Further experimental *in vivo* work using animal models will be needed to understand cause and

consequence of their regulation and their contribution to autistic phenotypes.

To conclude, the present PGAS work provides first evidence that a particular constellation of common genetic variants in the "broader fragile X gene family" contributes to autistic phenotypes. Even though still preliminary, the potential coordinator role of the miR-181 family seems highly interesting and worth pursuing, perhaps even with respect to future ASD treatment approaches.

## Materials and Methods

### Subjects

*Discovery sample (schizophrenia patients of the GRAS cohort)*
The Göttingen Research Association for Schizophrenia (GRAS) data collection has been established over the last 10 years and consists of > 1,200 deep phenotyped patients, diagnosed with

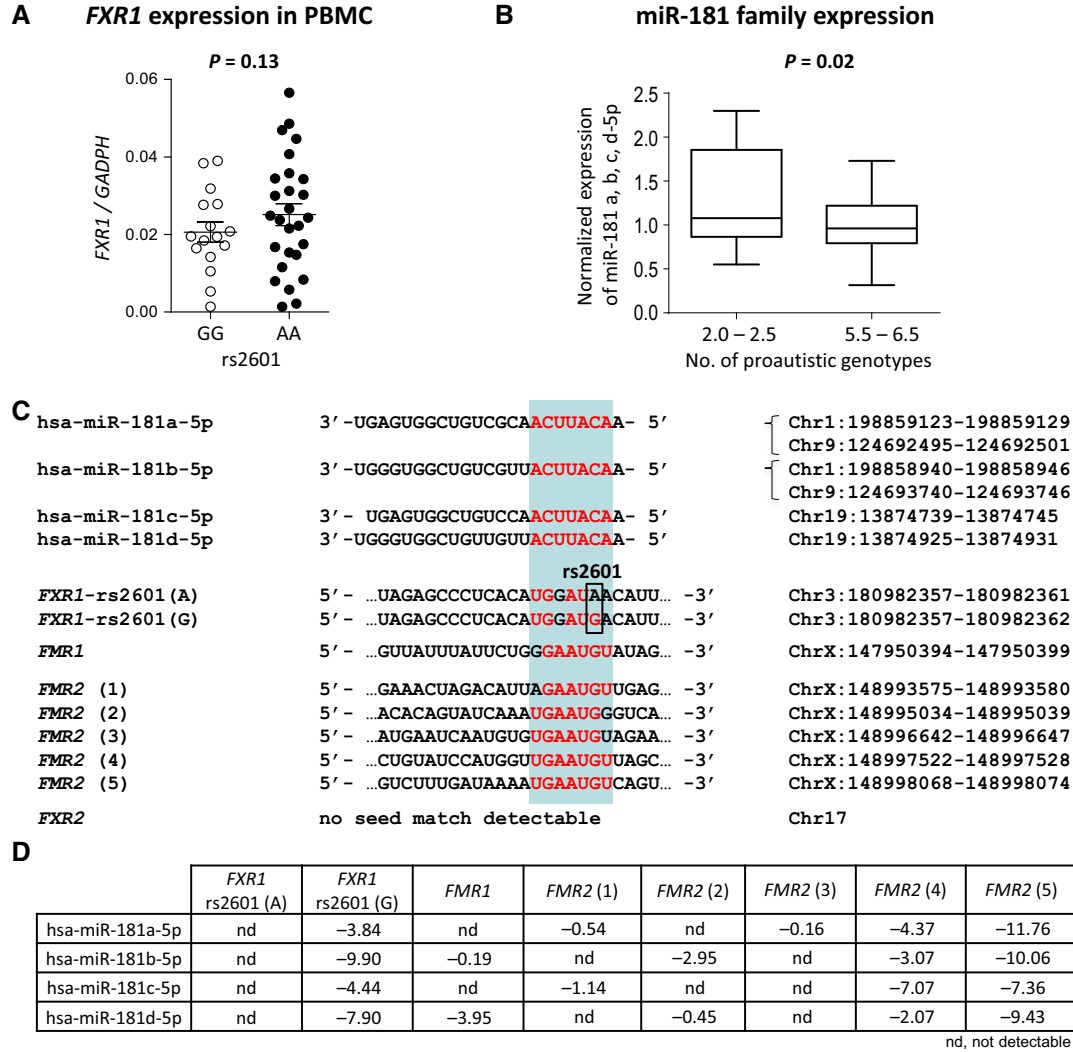

**Figure 5.   Pilot experiments toward first mechanistic insight.**

A    *FXR1* expression in PBMC of individuals carrying the rs2601 GG (low risk; *N* = 16) versus AA genotype (high risk; *N* = 27). Data represent mean ± SEM.
B    PBMC microRNA expression was normalized and data are plotted from all 4 miR-181-5p members (miR-181a, miR-181b, miR-181c, d miR-181-5p). The bottom and top of the box are the first and third quartiles; the band inside the box is the median; the ends of the whiskers represent minimum and maximum values of the data.
C    Sequence homology of all four human mature miR-181 species shown together with sequences in the broader fragile X gene family containing binding sites for the miR-181 family (seed matches identified using Target Scan Human and SFOLD). The red letters specify seed sequences and seed matches, respectively. Chromosome positions for each seed sequence and seed match are denoted (human genome assembly GRCh38/hg38).
D    Denoted are ΔΔG values for binding of each of the miR-181 family members to different 3'UTR positions. Positions were identified using Target Scan Human and SFOLD and then processed using PITA algorithm to yield the denoted ΔΔG values. ΔΔG is an energetic score, and the more negative its value, the stronger is the expected binding of the microRNA to the given site (Kertesz *et al*, 2007).

schizophrenia or schizoaffective disorder according to DSM-IV-TR criteria (American Psychiatric Association, 2000), recruited across 23 collaborating centers across Germany (Begemann *et al*, 2010; Ribbe *et al*, 2010). The study complies with the Helsinki Declaration and was approved by the Ethics Committee of the Georg-August-University (Göttingen, Germany) as well as all participating centers. All patients and/or their authorized legal representatives gave written informed consent. The present study focused on males only (*N* = 692 with complete data of a total of *N* = 739) since the male gender is more commonly affected by both, ASD and schizophrenia (Elsabbagh *et al*, 2012). Moreover, 2 of the

genes included in the accumulation model are X-chromosomal. The average age of the discovery sample was 37.29 ± 12.06 years (range 17–78).

*Healthy control sample (GRAS healthy blood donors)*
Healthy male controls for genetic case–control analysis were voluntary blood donors (*N* = 783; mean age 38.41 ± 13.29 years; range 18–69) from Department of Transfusion Medicine of the George-August-University (Göttingen, Germany) which widely fulfill health criteria, assessed by predonation screening (Begemann *et al*, 2010).

*Replication sample I (schizophrenia patients, Munich/Halle cohort)*
To replicate in an independent schizophrenia sample, the phenotype–genotype associations found in GRAS patients, data from male subjects ($N = 626$; mean age $35.49 \pm 11.22$ years; range 18–67) of the Munich/Halle collection, diagnosed with schizophrenia according to DSM-IV-TR criteria (American Psychiatric Association, 2000), were analyzed (Van den Oord *et al*, 2006). Written informed consent had been obtained from all subjects after detailed and extensive description of the study, which was approved by the local ethics committees and carried out according to the Declaration of Helsinki.

*Replication sample II (GRAS disease control cohort)*
Replication sample II consists of 111 males (mean age $42.37 \pm 14.87$ years; range 18–75) diagnosed with psychiatric disorders other than schizophrenia (71.2% affective disorders, 15.3% substance use disorders, 13.5% others) according to DSM-IV-TR (American Psychiatric Association 2000). All patients were originally recruited as part of the GRAS study but upon careful diagnosis validation (review of psychiatric history, Structured Clinical Interview for DSM-IV, SCID I) failed to fulfill DSM-IV-TR criteria for schizophrenia or schizoaffective disorder.

*Replication sample III (population-based cohort, SHIP)*
Replication sample III comprises 2,005 male subjects (mean age $50.91 \pm 16.41$ years; range = 20–80) from the baseline cohort of the Study of Health in Pomerania (SHIP), conducted in North-East Germany. SHIP investigates common risk factors and subclinical disorders and manifests diseases in the general population (Völzke *et al*, 2011). All participants gave written informed consent. The survey and study methods were approved by the institutional review boards of the University of Greifswald.

### Phenotyping

*Target phenotype: the PANSS autism severity score (PAUSS)*
To capture autistic features, an autism severity score (PAUSS) was calculated for the discovery sample and replication samples I and II (Kästner *et al*, 2015). It represents the mean of six items of the negative and two items of the general subscale of the Positive and Negative Syndrome Scale (PANSS) (Kay *et al*, 1987). For the replication sample III, the Instrumental Support Index (ISI, self-report) was used as proxy phenotype indicating the quality of instrumental and emotional support (Klein *et al*, 2012). The ability to establish and maintain high-quality relationships is crucial for receiving social support when needed. These skills are expected to be low in individuals with ASD or with strong autistic phenotypes and should be reflected by lower support. To cross-validate this proxy phenotype in the discovery sample (GRAS), social support was operationalized as the self-reported number of individuals a person can rely on in case of emergency. For both measures of social support, higher values represent higher social support, that is, lower autistic features.

*Further schizophrenia-relevant phenotypes*
For the discovery sample, the severity of psychotic symptoms was evaluated by the PANSS (Kay *et al*, 1987). The Cambridge Neurological Inventory (CNI) (Chen *et al*, 1995) was applied as a measure

of neurological functioning and a number of neuropsychological tests were administered. As global measure of cognitive functioning, a cognitive composite score, comprising reasoning (subtest 3, Leistungsprüfsystem, [LPS3]), executive function (Trail-Making Test, part B [TMT-B], and verbal learning and memory (Verbal Learning and Memory Test [VLMT]), was employed (Begemann *et al*, 2010). As control phenotype, a delusional-depression composite score was used, based on other GRAS database items (Fig 3C) (Ribbe *et al*, 2010).

### Microsatellite analysis

For the discovery sample and the healthy control sample (GRAS), 2 polymorphic repeats in the 5′UTR regions of *FMR1* and *FMR2* genes were amplified from genomic DNA by PCR. Primers for CGG/CCG repeats at the *FMR1/FMR2* genes were designed according to published information (Fu *et al*, 1991; Santos *et al*, 2001).

*FMR1: CGG repeat*
Primer c: 5′-GCTCAGCTCCGTTTCGGTTTCACTTCCGGT-3′ (labeling VIC)
Primer f: 5′-AGCCCCGCACTTCCACCACCAGCTCCTCCA-3′

*FMR2: CCG repeat*
Primer XE2: 5′-GCCCTCCCGCCCAGCTAAAAGTGTCCGGG-3′ (labeling FAM)
Primer 603: 5′-CCTGTGAGTGTGTAAGTGTGTGATGCTGCCG-3′

For each sample, the reaction mixture (21 μl) was prepared in 384 well plates, each containing 1 μl (50 ng) of human genomic DNA, 10 μl QIAGEN Multiplex PCR Plus Kit (Qiagen, Hilden, Germany), 6 μl Q-Solution (Qiagen) and 4 μl VIC- or FAM-labeled forward primer and unlabeled reverse primer (190 nM final each). The cycling program was carried out after a preheating step at 98°C for 5 min and included 35 cycles of: (1) denaturation at 98°C for 45s, (2) annealing at 68°C for 2 min and (3) extension at 72°C for 2 min, followed by a final extension step at 72°C for 20 min in a DNA Thermal Cycler (PTC-200 MJ Research, Bio-Rad, Munich, Germany). The amplicons were separated using size electrophoresis on the ABI 3730 XL DNA Analyzer. Samples were diluted 1:50 with 0.3 mM EDTA and 4 μl was mixed with 6 μl LIZ-500 Size Standard (Applied Biosystems, Foster City, CA, USA). Raw data were processed using the Gene Mapper Software 4.0 (Applied Biosystems). Overall, successfully genotyped markers amounted to 98.1%.

### Genotyping

*GRAS sample*
The GRAS sample (discovery sample, replication sample II—disease controls, and healthy controls) was genotyped using a semicustom Axiom MyDesign Genotyping Array (Affymetrix, Santa Clara, CA, USA), based on a CEU (Caucasian residents of European ancestry from UT, USA) marker backbone including 518,722 SNPs, and a custom marker set including 102,537 SNPs. Genotyping was performed by Affymetrix on a GeneTitan platform. Several quality control steps were applied (SNP call rate > 97%, Fisher's linear discriminant, heterozygous cluster strength offset, homozygote ratio

offset) (Hammer *et al*, 2014; Stepniak *et al*, 2014). From the 739 GRAS males, 30 individuals had to be excluded from further analysis due to relatedness, genotyping problems, and genetic population outlier status (based on 10 principle components). Similarly, of the 783 healthy controls, 53 individuals had to be excluded from further analysis for the same reasons.

### Replication sample I

Replication sample I was genotyped using the iPLEX assay on the MassARRAY MALDI-TOF mass spectrometer as described (Oeth *et al*, 2009). Allele-specific extension products were identified and genotypes allocated by Typer 3.4 Software (Sequenom, San Diego, CA, USA). All applied quality criteria were met (individual call rate > 80%, SNP call rate > 99%, identity of genotyped CEU Trios [Coriell Institute for Medical Research, Camden, NJ] with HapMap database > 99%).

### Replication sample III—SHIP-0

Replication sample III, SHIP-0, was genotyped using the Affymetrix Genome-Wide Human SNP Array 6.0. Hybridization of genomic DNA was done in accordance with the manufacturer's standard recommendations. The overall genotyping efficiency was 98.6%. Imputation of genotypes in the SHIP cohort was performed with the software IMPUTE v0.5.0 against the 1,000 Genomes (phase 1v3) reference panel using 869,224 genotyped SNPs (Völzke *et al*, 2011).

### Marker selection

From the broader fragile X gene family (*FMR1, FXR1, FXR2, FMR2*), markers were in a first step preselected according to the following selection criteria: (i) SNPs with potentially functional significance (located in promoter region, 3′UTR or coding sequence) to later facilitate potential mechanistic insight; (ii) SNPs with reasonable minor allele frequency (MAF ≥ 0.2) to allow for statistical analyses in our discovery sample; and (iii) SNPs not in high linkage disequilibrium (LD) with another selected SNP ($r^2 < 0.8$) to exclude redundant information (priority was given here to the SNPs fulfilling the aforementioned criteria of functionality and MAF). This preselection yielded a total of 13 SNPs (Fig 2). In a second step, a phenotype-based genetic association study (PGAS) approach using the PAUSS (Kästner *et al*, 2015) was performed individually on all 13 SNPs. SNPs with a tendency of one genotype being associated with an autistic phenotype went into the final accumulation model of 8 SNPs (standard operating procedure, SOP, explained in Figs 2 and 3). Not more than 3 SNPs per gene were ultimately selected to avoid over-representation of one gene (Figs 1A, 2 and 3).

### Isolation of peripheral blood mononuclear cells (PBMC)

Morning blood samples were obtained via phlebotomy into CPDA vials (Citrate Phosphate Dextrose Adenine, Sarstedt, Germany). PBMC were isolated applying the standard Ficoll-Paque Plus isolation procedure (GE Healthcare, Munich, Germany).

### Reverse transcription and real-time PCR

Total RNA was extracted with RNAeasy kit (Qiagen, Hilden, Germany) from PBMC and cDNA synthesis carried out via SuperscriptIII kit (Invitrogen, Karlsruhe, Germany). *FXR1* and *GAPDH* cDNA was detected in LightCycler480 via SYBR green (Roche, Diagnostics GmbH, Mannheim, Germany) using specific primers:

*FXR1*: 5′- AGAGGCCACTAAGCATTTAGAA-3′ (forward)
*FXR1*: 5′-TCTCCGTAGATTCTGAATGTTCCA-3′ (reverse)
*GAPDH*: 5′-CTGACTTCAACAGCGACACC-3′ (forward)
*GAPDH*: 5′-TGCTGTAGCCAAATTCGTTGT-3′ (reverse)

### Small RNA sequencing

RNA was extracted from PBMC using TRI Reagent, chloroform, and isopropyl alcohol. Library preparation and cluster generation was performed according to the Illumina standard protocols (TruSeq, Illumina, San Diego, CA, USA). Libraries were quality-controlled and quantified using a Nanodrop 2000 (Thermo Scientific, Darmstadt, Germany), Agilent 2100 Bioanalyzer (Agilent Technologies, Waldbronn, Germany) and Qubit (Life Technologies, Darmstadt, Germany). Base calling from raw images and file conversion to fastq files were achieved by Illumina pipeline scripts. 3′ adapters were trimmed and filtered for reads with the minimum length of 15 nucleotides using cutadapt (Martin, 2011) prior to the mapping. Reads were then mapped to a reference genome created from the mature microRNA sequences in human genome using RNA-STAR (Dobin *et al*, 2013). No mismatches for reads < 20 b were allowed, 1 mismatch was allowed for reads between 20 b and 39 b, and 2 mismatches for reads between 40 b and 59 b. Reads were mapped in non-splice-junction-aware mode. Remaining unmapped reads were then mapped to the human genome (GRC37), and only high-quality reads (MAPQ ≥ 30) were considered. In order to obtain the total number of uniquely mapped reads for each sample, high-quality uniquely mapped reads from mature microRNA and human genome (GRC37) were combined. The normalized read counts were then obtained by dividing the read's count by the total number of uniquely mapped reads for the samples, respectively. Since RNA sequencing is not biased by probe design, we were able to quantify miR-181 levels using the normalized read counts for miR-181a, miR-181b, miR-181c, and miR-181d-5p detected in high- and low-risk groups for statistical analysis.

### FMRP determination

PBMC were resuspended in lysis buffer containing 1% NP-40 in PBS (140 mM NaCl, 2.7 mM KCl, 3.2 mM $Na_2HPO_4$, 1.5 mM $KH_2PO_4$, pH 7.4) and incubated for 10 min on ice. Lysates were cleared by centrifugation at 11,200 *g* for 10 min at 4°C and protein concentrations determined by Bradford colorimetry (Bio-Rad City, CA, USA). Extracts containing 60 μg total protein were separated by SDS–PAGE and transferred to a polyvinylidene difluoride membrane (Millipore, Schwalbach, Germany) by Western blotting. The membrane was blocked with 5% w/v nonfat milk in TBT (150 mM NaCl, 6 mM Tris base, 15 mM Tris–HCl pH 7.5, 0.5% Tween-20) at room temperature for 40 min and incubated with the primary antibodies against FMRP (1:200, Millipore MAB2160) and beta-actin (house keeper; 1:1,000, Sigma Aldrich A5316, Seelze, Germany) at 4°C for 12 h. After washing with TBT, the membrane was incubated with a horseradish peroxidase-conjugated anti-mouse antibody

**The paper explained**

**Problem**

Fragile X syndrome (FXS) is caused by a mutation, mostly a large CGG triplet expansion in the regulatory region of the fragile X mental retardation 1 gene (*FMR1*). Up to 60% of males with FXS fulfill criteria for autism spectrum disorder (ASD), making FXS the most common single-gene cause of syndromic ASD. Since FXS is a X-chromosomal disorder, males are generally more severely affected. In the present study, we asked for the first time whether accumulated normal genetic variants in genes of the fragile X family (*FMR1, FXR1, FXR2*), to which we added here *FMR2* due to its prominent phenotypical similarities, modulate autistic features in males, independently of the described mutations.

**Results**

We report here an accumulation model of 8 common single nucleotide polymorphisms from the "broader fragile X family" (*FMR1, FXR1, FXR2, FMR2*) that yields significant association with autistic traits in a discovery sample of male patients with schizophrenia (*N* = 692) and in three independent male replicate samples: (I) Patients with schizophrenia from Munich/Halle (*N* = 626), (II) patients with psychiatric diagnoses other than schizophrenia (extended GRAS data collection; *N* = 111), and (III) a general population sample from Greifswald (*N* = 2,005). Searching for first mechanistic insight, we performed small RNA sequencing to contrast microRNA expression in peripheral blood mononuclear cells of extreme group subjects with high versus low autism severity score/genetic risk constellation. Interestingly, we found differential expression of the miR-181 family of brain-expressed microRNAs which have several seed matches across the "broader fragile X gene family".

**Impact**

Our data provide first evidence that a particular constellation of completely normal genotypes in the "broader fragile X gene family" contributes to autistic phenotypes. Linking unfortunate normality to disease should act in the sense of anti-stigma and supports the view of a continuum of autistic traits across health and disease. The miR-181 family seems to fulfill an "umbrella regulator" task and may be an interesting target for future modulatory treatments in ASD.

(1:5,000, Sigma Aldrich A4416) for 2 h at room temperature. The blot was developed with the enhanced chemiluminescence (ECL) system and signals were quantified (relative units) using ImageJ software (Rasband, W.S., ImageJ, U. S. National Institutes of Health, Bethesda, Maryland, USA, http://imagej.nih.gov/ij/, 1997–2014).

**Statistical analysis**

Mann–Whitney *U*-test or binary logistic regression was used for group comparisons or *t*-test in case of normally distributed dependent variables and no lack of homogeneity of variances. Spearman rank correlation coefficient was used to assess the strength of association between 2 non-parametric variables. Cronbach's alpha was calculated as a measure of internal consistency. For exclusion of statistical outliers, the Grubbs' test was used. Statistical analyses were performed with SPSS for windows version 17.0 (IBM-Deutschland GmbH, Munich, Germany) or with STATA/MP software, version 13 (StataCorp LP, College Station, TX). Case–control analysis of SNP genotypes as well as test for deviation from Hardy–Weinberg equilibrium was performed using PLINK 1.07 (Purcell *et al*, 2007).

**Acknowledgements**

This work was supported by the Max Planck Society, the Max Planck Förderstiftung, the DFG (CNMPB; project Psycourse), the BMBF funded eMED project Intergrament, EXTRABRAIN EU-FP7 and EU-AIMS. The research of EU-AIMS receives support from the Innovative Medicines Initiative Joint Undertaking under grant agreement no. 115300, resources of which are composed of financial contribution from the European Union's Seventh Framework Programme (FP7/2007-2013), from the EFPIA Companies, and from Autism Speaks. SHIP is part of the Community Medicine Research net of the University of Greifswald, which is funded by the Federal Ministry of Education and Research (grants no. 01ZZ9603, 01ZZ0103, and 01ZZ0403), the Ministry of Cultural Affairs and the Social Ministry of the Federal State of Mecklenburg-West Pomerania. Genome-wide data in SHIP have been supported by a joint grant from Siemens Healthcare, Erlangen, and the Federal State of Mecklenburg-West Pomerania. The authors thank all subjects for participating in the study, and all the many colleagues who have contributed over the past 10 years to the GRAS data collection.

**Author contributions**

HE conceptualized, planned, supervised, and coordinated the project. MB recruited and examined patients of the GRAS data collection. BS, MM, and AK were instrumental for all the GRAS database work. BS, AK, MM, and SVdA performed the PGAS study, including phenotypical and statistical analyses of discovery sample and replicates. MM, AH, DKB, and FB were involved in the genetic work-up of samples. GP, with the help of CBr, GM, and FB performed all laboratory work, except for the small RNA sequencing which was conducted and analyzed by FS and AF. MB, DKB, CBa, UF, AD, HJG, DR, and AF were instrumental for aspects of study design related to their expertise and respective data interpretation. DR, HJG, GH, and HV enabled the replication studies employing subjects from Halle/Munich and Greifswald. HE wrote the paper and supported by BS, AK, and GP. All authors read and approved the final version of the manuscript.

**Conflict of interest**

The authors declare that they have no conflict of interest.

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
