## [Review Process File · EMBO Molecular Medicine]

Accumulated common variants in the broader fragile X gene family modulate autistic phenotypes

Beata Stepniak, Anne Kästner, Giulia Poggi, Marina Mitjans, Martin Begemann, Annette Hartmann, Sandra Van der Auwera, Farahnaz Sananbenesi, Dilja Krueger-Burg, Gabriela Matuszko, Cornelia Brosi, Georg Homuth, Henry Völzke, Fritz Benseler, Claudia Bagni, Utz Fischer, Alexander Dityatev, Hans-Jürgen Grabe, Dan Rujescu, Andre Fischer, and Hannelore Ehrenreich

Corresponding author: Hannelore Ehrenreich, Max Planck Institute of Experimental Medicine

Review timeline:

Submission date:	30 July 2015
Editorial Decision:	14 October 2015
Revision received:	21 October 2015
Editorial Decision:	26 October 2015
Accepted:	26 October 2015

Transaction Report:

Editor: Roberto Buccione

1st Editorial Decision

14 October 2015

Thank you for the submission of your manuscript to EMBO Molecular Medicine. We have now heard back from three Reviewers whom we asked to evaluate your manuscript.

We are very sorry that it has taken so long to get back to you on your manuscript. In fact, as I had previously mentioned, we experienced unusual difficulties in securing three willing and appropriate reviewers and in obtaining their evaluations in a timely manner and finally, we were also unable to further contact a reviewer (#2), which forced us to search for a new one. Hopefully the inevitable frustration due to this delay will be somewhat tempered by the generally positive evaluations.

You will see that all three Reviewers are supportive of your work, although Reviewers 3 and 4, are specifically critical of the lack of mechanistic development concerning the relationship between the miR-181 family and the fragile X family.

Reviewer 1 would like you to better discuss and contextualize your results with respect to the available knowledge. Reviewer 4 lists a number of specific and well-taken items for your action

In conclusion, while publication of the paper cannot be considered at this stage, given the potential interest of your findings, we would be pleased to consider a revised submission, with the understanding that the Reviewers' concerns must be addressed with additional experimental data where appropriate and that acceptance of the manuscript will entail a second round of review. Please note that after further discussion with my colleagues and the reviewers, I will not be requiring you to address the mechanistic concerns directly (provided all other issues are carefully and fully dealt

with). I do, however encourage you to develop your study as far as realistically possible in a mechanistic sense for your next, revised version to strengthen your findings and increase their impact. Where unsubstantiated, I would advise toning down your conclusions.

We wish to remind you that it is EMBO Molecular Medicine policy to allow a single round of revision only and that, therefore, acceptance or rejection of the manuscript will depend on the completeness of your responses included in the next, final version of the manuscript.

As you know, EMBO Molecular Medicine has a "scooping protection" policy, whereby similar findings that are published by others during review or revision are not a criterion for rejection. I understand that in this case, to address the above might entail a significant amount of additional work and time and might be technically challenging. However, I do ask you to get in touch with us after three months if you have not completed your revision, to update us on the status. Please also contact us as soon as possible if similar work is published elsewhere.

Please note that EMBO Molecular Medicine now requires a complete author checklist (<http://embomolmed.embopress.org/authorguide#editorial3>) to be submitted with all revised manuscripts. Provision of the author checklist is mandatory at revision stage; The checklist is designed to enhance and standardize reporting of key information in research papers and to support reanalysis and repetition of experiments by the community. The list covers key information for figure panels and captions and focuses on statistics, the reporting of reagents, animal models and human subject-derived data, as well as guidance to optimise data accessibility.

Last, but not least, please carefully conform to our author guidelines (<http://embomolmed.embopress.org/authorguide>; see also further below) to ensure rapid pre-acceptance processing in case of a favourable outcome on your revision.

I look forward to seeing a revised form of your manuscript as soon as possible.

***** Reviewer's comments *****

Referee #1 (Remarks):

While this is an interesting study of the relevance of variants of the fragile X gene family in non - autistic samples, it would have been more informative to include discussion of results by other labs showing relevance of abnormalities in FMRP levels in schizophrenic, and mood disorder samples (Kelemen et al. 2013; Kovacs et al. 2013; Fatemi et al. 2010) as well a mechanistic explanation of important role of FMRP in brain function and synaptic plasticity (Darnell et al. 2011).

Referee #3 (Comments on Novelty/Model System):

The authors started to develop a model to explain the findings by determining Micro RNA Expression in blood cells. This is the starting Point of further experimental approaches that have not been performed according to the manuscript.

Referee #3 (Remarks):

Manuscript: "Accumulated common variants in the broader fragile X gene family modulate autistic phenotypes" by Szepianiak et al..

The paper by Szepianiak et al. deals with the question if the "broader fragile X family" of genes that might be -if mutated or specifically altered (normal genetic variant)- also be responsible or predictive for autism spectrum disorders (ASDs). To that end the authors analysed a discovery sample of schizophrenic patients and 3 independent replicate samples (schizophrenia, psychiatric diagnosis, normal population). The authors found an accumulation of 8 common single nucleotide polymorphisms in the FMR1, FXR1, FXR2 and FMR2 genes. Since there is a common putative

regulation site within these genes with hsa-miR-181 microRNAs the authors determined the expression of these RNAs in patient blood monocytes.

The paper describes an interesting finding connecting autistic like features to single nucleotide polymorphisms within the family of fragile X genes. The methods described are sound and the data provided are novel.

The authors made a first step for answering the question how these sequence changes might explain molecular alterations. Here they started to analyse the impact of microRNAs on the expression of genes of the broader fragile X family. The results are a hint to a role of miR-181 but the data provided are very preliminary and urgently need further analysis in the test tube.

Referee #4 (Comments on Novelty/Model System):

1. Autism spectrum disorders are heterogeneous and complex. Current human genetics has revealed rare variants that can only explain a minority of autism cases. Multiple common variants almost certainly play an important role, but a firm grasp on the genetics of common variants is lacking. This paper is original in this respect and contains elements that can be pivotal in this area.
2. The methodology is innovative: genetic association based on phenotypes. The way the authors apply this to the problem is of great interest to the field.

Referee #4 (Remarks):

This manuscript focusses on the connection of four fragile-X-related genes with autism phenotypes. It contains two parts: (i) the correlation between the occurrence of common variants in these genes and phenotypes, and (ii) a further correlation with a micro-RNA matches and levels (miR-181 family). The paper implicates common variation in the FX-related genes in the severity of autistic symptoms.

Strong points:

1. Autism spectrum disorders are heterogeneous and complex. Current human genetics has revealed rare variants that can only explain a minority of autism cases. Multiple common variants almost certainly play an important role, but a firm grasp on the genetics of common variants is lacking. This paper is original in this respect and contains elements that can be pivotal in this area.
2. The methodology is innovative: genetic association based on phenotypes. The way the authors apply this to the problem is of great interest to the field.
3. The studies have been carried out with a very suitable set of cohorts.

Weak / unclear points:

4. There is no page numbering, which makes it hard to refer precisely to the text.
5. The selection of eight 'pro-autistic genotypes' (Results) from 13 SNPs lacks criteria: how was this achieved, were combinations of genotypes systematically optimized to the tendency to associate with a higher PAUSS? The data obtained in this process should be presented in Supplementals.
6. What are homozygous autosomal pro-autistic genotypes: again, criteria and background data are lacking.
7. It seems logical that assignment of pro-autistic genotypes correlates with higher PAUSS scores. The presented data do not convince that there is specificity for the FX genes.
8. The concept of 'umbrella regulator' should be specified. What is precisely meant? And, how see the authors as mechanistic link between the given genomic variations and the observed changes in miR-181 family RNAs? Is cause and consequence assumed. If so, at what level; if not, what biological mechanism drives the correlation? This part seems immature.

Taken together, the manuscript has significant merits in concept and approach, but lacks sufficient clarity and biological embedding. It can gain with revisions and additions.

EMM-2015-05696

POINT-BY-POINT RESPONSE TO THE REVIEWERS' COMMENTS

(Original reviewers' comments in black; our response in blue; reference to respective passages/ changes/ additions in the manuscript in yellow)

General Response:

In addition to the point-by-point response below and the herewith associated changes in the manuscript, we decided – inspired by the reviewers' comments - to make our innovative phenotype-based genetic association study (PGAS) approach even clearer in the revised manuscript. Instead of hiding our standard operating procedure (SOP) for PGAS in a supplement, we integrated now all tables and figures into the main manuscript. Moreover, we provide an entirely new figure (new Figure 3) and a new table (Table IV). This will certainly help in the future to avoid misunderstandings as they appeared in the present review process. Moreover, as suggested by Referee #4 (*"The methodology is innovative: genetic association based on phenotypes. The way the authors apply this to the problem is of great interest to the field"*), our SOP may serve as a guideline for the interested scientific community to perform similar approaches.

Referee #1 (Remarks):

While this is an interesting study of the relevance of variants of the fragile X gene family in non -autistic samples, it would have been more informative to include discussion of results by other labs showing relevance of abnormalities in FMRP levels in schizophrenic, and mood disorder samples (Kelemen et al, 2013; Kovacs et al. 2013; Fatemi et al. 2010) as well a mechanistic explanation of important role of FMRP in brain function and synaptic plasticity (Darnell et al. 2011).

We thank the referee for taking time to review our manuscript.

We have to clarify an **important misunderstanding**: We did **not** really work on FMRP in this manuscript. Our focus was on genotype-phenotype associations including 8 normal variants in 4 genes of the fragile X family, among them *FMR1*. As a starting point, we made sure that all of our discovery sample patients are in both *FMR1* and *FMR2* way below the repeat polymorphism length relevant for any genetically induced alterations in the respective RNA binding protein levels (Figure 1). Most importantly, we replicate our genotype-phenotype associations with the 8SNP model also in healthy individuals. The references mentioned by this reviewer (Kelemen et al 2013; Kovacs et al 2013; Fatemi et al 2010) reported on FMRP levels and schizophrenia; unfortunately, in these papers the essential control of the repeat length has not been done. All analyses are only based on ELISA or Western blot. Therefore, even though having been aware of these papers and the letter, we did not cite them since this would be confusing and out of the focus of the present manuscript.

We did, however, at some point even quantify FMRP in our extreme groups. Here are the results for the reviewer's interest:

Comparison of FMRP levels between extreme groups based on number of risk genotypes. Western blot quantification of FMRP expression in peripheral blood mononuclear cells of male GRAS patients is shown. LR=low risk (1.5-2.5 risk genotypes). HR=high risk (5.5-6.5 risk genotypes). The bars display the densitometric quantification data that reveal no significant differences in the relative amount of FMRP between both extreme groups (all with known normal repeat lengths). For group comparison, the *t* test was used: 1.5-2.5 (N=6) versus 5.5-6.5 (N=7) risk genotypes: 0.418 ± 0.493 versus 0.522 ± 0.651 , mean \pm SEM; $p=0.754$. This information has now been added to the text, page 9 and to materials & methods, page 17.

The suggested very interesting reference Darnell et al. 2011 has now been cited (page 3) even though our paper is not on FMRP.

Referee #3 (Comments on Novelty/Model System):

The authors started to develop a model to explain the findings by determining Micro RNA Expression in blood cells. This is the starting Point of further experimental approaches that have not been performed according to the manuscript.

Yes, indeed! The reviewer is right. Our observation of the miR-181 family showing different expression in PBMC of extreme groups with high versus low autism severity score are an interesting starting point for further work. This the more so since our screening for the common miR-181 seed sequence revealed a surprising number of miR-181 seed matches in the broader fragile X gene family (see Figure 5). As suggested by this reviewer, we are now planning to perform mechanistic experiments on the role of miR-181 in the context of our results. However, this is a project by itself that involves *in vivo* manipulations of miR-181 in mice and research in IPS cells and is thus certainly beyond the scope of the present manuscript.

Referee #3 (Remarks):

Manuscript: "Accumulated common variants in te broader fragileX gene family modulate autistic phenotypes" by Szephaniak et al..

The paper by Stephaniak et al. deals with the question if the "broader fragile X family" of genes that might be -if mutated or specifically altered (normal genetic variant)- also be responsible or predictive for autism spectrum disorders (ASDs). To that end the authors analysed a discovery sample of schizophrenic patients and 3 independent replicate samples (schizophrenia, psychiatric diagnosis, normal population). The authors found an accumulation of 8 common single nucleotides polymorphisms in the FMR1, FXR1, FXR2 and FMR2 genes. Since there is a common putative regulation site within these genes with hsa-miR-181 microRNAs, the authors determined the expression of these RNAs in patient blood monocytes. The paper describes an interesting finding connecting autistic like features to single nucleotide polymorphisms within the family of fragile X genes. The methods described are sound and the data provided are novel.

We thank the reviewer for reading our manuscript and for the positive comments.

The authors made a first step for answering the question how these sequence changes might explain molecular alterations. Here they started to analyse the impact of microRNAs on the expression of genes of the broader fragile X family. The results might hint to a role of miR-181 but the data provided are very preliminary and urgently need further analysis in the test tube.

We agree with the reviewer that the data on the miR-181 family are an unexpected discovery that needs now further work in animal models and human IPS-derived neurons. This work is actually planned to be performed in our labs over the next 3-5 years, and a respective grant proposal will be submitted upon acceptance of this paper. See also response above.

The present work has focused on the development and validation of the 8SNP model based on a phenotype-based genetic association (PGAS) approach, including several replications (this becomes now clearer in the revised manuscript), and ends with the interesting preliminary finding of a regulation of the miR-181 family. This finding provoked the hypothesis that the miR-181 family might play the role of an 'umbrella regulator' for the fragile X family of genes and perhaps also other autism relevant genes. By toning down some of our wording, we made now sure that the still hypothetical character of this preliminary conclusion becomes better transparent. Please see throughout the manuscript.

Referee #4 (Comments on Novelty/Model System):

1. Autism spectrum disorders are heterogeneous and complex. Current human genetics has revealed rare variants that can only explain a minority of autism cases. Multiple common variants almost certainly play an important role, but a firm grasp on the genetics of common variants is lacking. This paper is original in this respect and contains elements that can be pivotal in this area.
2. The methodology is innovative: genetic association based on phenotypes. The way the authors apply this to the problem is of great interest to the field.

We thank the reviewer for these very stimulating and encouraging comments. Based on these positive remarks, we have now integrated and explained the used novel methodology even better in the revised manuscript. See the new Table IV, and the new Figure 3 including legends.

Referee #4 (Remarks):

This manuscript focusses on the connection of four fragile-X-related genes with autism phenotypes. It contains two parts: (i) the correlation between the occurrence of common variants in these genes and phenotypes, and (ii) a further correlation with a micro-RNA matches and levels (miR-181 family). The paper implicates common variation in the FX-related genes in the severity of autistic symptoms.

In the revised manuscript, we have now made it clearer that part 1 of the manuscript is the main part and part 2 is still preliminary and will be the basis of future work. Please see throughout the text.

Strong points:

1. Autism spectrum disorders are heterogeneous and complex. Current human genetics has revealed rare variants that can only explain a minority of autism cases. Multiple common variants almost certainly play an important role, but a firm grasp on the genetics of common variants is lacking. This paper is original in this respect and contains elements that can be pivotal in this area.
2. The methodology is innovative: genetic association based on phenotypes. The way the authors apply this to the problem is of great interest to the field.
3. The studies have been carried out with a very suitable set of cohorts.

We thank this referee for the positive and encouraging feedback.

Weak / unclear points:

4. There is no page numbering, which makes it hard to refer precisely to the text.

We have to apologize. We are very sorry for the inconvenience – this has indeed escaped our attention.... Page numbering has now been included.

5. The selection of eight 'pro-autistic genotypes' (Results) from 13 SNPs lacks criteria: how was this achieved, were combinations of genotypes systematically optimized to the tendency to associate with a higher PAUSS? The data obtained in this process should be presented in Supplementals.

We thank the Reviewer for pointing this out. Inspired by these comments, we decided to make our innovative PGAS approach even clearer in the revised manuscript. Instead of hiding our standard operating procedure (SOP) in a supplement, we integrated now all tables and figures into the main manuscript. This may help to avoid misunderstandings as they appeared in this review process. Moreover, as indirectly suggested by this Referee, it may serve as a guideline for the interested scientific community to perform similar approaches. See in the revised manuscript the new Table IV and the new Figure 3.

6. What are homozygous autosomal pro-autistic genotypes: again, criteria and background data are lacking.

This has now been explained in great detail in the new Figure 3 plus legend and also in the main text, page 6.

7. It seems logical that assignment of pro-autistic genotypes correlates with higher PAUSS scores. The presented data do not convince that there is specificity for the FX genes.

The answer may be given in 2 parts: (1) Specificity of the association of our 8SNP model with the autistic phenotype (PAUSS) is now demonstrated in Figure 3 C and D, using a control phenotype, the delusional-depression score. (2) On the other hand, we would like to emphasize that there is no exclusivity of the fragile X genes for determining autistic features. They do, however, play an interesting, and across groups (health and disease) validated, modulator role. Many other genes that were not subject of the present work, additionally act as important modifiers of autistic phenotypes. See paper, page 10.

8. The concept of 'umbrella regulator' should be specified. What is precisely meant?

This term is used to describe a potential regulatory principle that is common to different independent genes (that are even located on different chromosomes) that all carry seed sequences of the miR-181 species which therefore can influence their expression in the sense of an overarching regulatory mechanism. This explanation has now been added to page 11.

And, how see the authors as mechanistic link between the given genomic variations and the observed changes in miR-181 family RNAs? Is cause and consequence assumed. If so, at what level; if not, what biological mechanism drives the correlation? This part seems immature.

We agree that these questions remain open for the moment. Therefore, we only state: A principle risk constellation may be given if miR-181 binding is reduced (as in the 3'UTR SNP rs2601 AA), or if miR-181 levels are decreased (expression diminished or consumption/degradation increased), together resulting as net effect in less efficient downregulation of target genes... see page 11. In addition, we have now made the lack of insight into cause and consequence clearer. See page 11.

Taken together, the manuscript has significant merits in concept and approach, but lacks sufficient clarity and biological embedding. It can gain with revisions and additions.

These revisions have now been performed and additions included.

2nd Editorial Decision

26 October 2015

Please find enclosed the final reports on your manuscript from the reviewer whom we asked to re-evaluate it. We are pleased to inform you that your manuscript is accepted for publication and is being sent to our publisher to be included in the next available issue of EMBO Molecular Medicine.

Congratulations on your interesting work,

***** Reviewer's comments *****

Referee #4 (Remarks):

The authors was appropriately revised the manuscript within the limits of the current study.